# Diagnosis of Schistosomiasis without a Microscope: Evaluating Circulating Antigen (CCA, CAA) and DNA Detection Methods on Banked Samples of a Community-Based Survey from DR Congo

**DOI:** 10.3390/tropicalmed7100315

**Published:** 2022-10-19

**Authors:** Pytsje T. Hoekstra, Joule Madinga, Pascal Lutumba, Rebecca van Grootveld, Eric A. T. Brienen, Paul L. A. M. Corstjens, Govert J. van Dam, Katja Polman, Lisette van Lieshout

**Affiliations:** 1Department of Parasitology, Leiden University Medical Center, 2333 ZA Leiden, The Netherlands; 2Institute of Health and Society, Université Catholique de Louvain, 1348 Brussels, Belgium; 3Department of Biomedical Sciences, Institute of Tropical Medicine, 2000 Antwerp, Belgium; 4Institut National de Recherche Biomédicale, Kinshasa 1197, Democratic Republic of the Congo; 5Department of Tropical Medicine, University of Kinshasa, Kinshasa 7948, Democratic Republic of the Congo; 6Department of Clinical Microbiology, Leiden University Medical Center, 2333 ZA Leiden, The Netherlands; 7Department of Cell and Chemical Biology, Leiden University Medical Center, 2333 ZA Leiden, The Netherlands; 8Department of Health Sciences, VU University Amsterdam, 1081 HV Amsterdam, The Netherlands

**Keywords:** schistosomiasis, diagnostics, community-based survey, circulating antigen, DNA detection, CCA, CAA, PCR, urine, stool

## Abstract

Detection of *Schistosoma* eggs in stool or urine is known for its low sensitivity in diagnosing light infections. Alternative diagnostics with better sensitivity while remaining highly specific, such as real-time PCR and circulating antigen detection, are progressively used as complementary diagnostic procedures but have not yet replaced microscopy. This study evaluates these alternative methods for the detection of *Schistosoma* infections in the absence of microscopy. Schistosomiasis presence was determined retrospectively in 314 banked stool and urine samples, available from a previous survey on the prevalence of taeniasis in a community in the Democratic Republic of the Congo, using real-time PCR, the point-of-care circulating cathodic antigen (POC-CCA) test, as well as the up-converting particle lateral flow circulating anodic antigen (UCP-LF CAA) test. *Schistosoma* DNA was present in urine (3%) and stool (28%) samples, while CCA (28%) and CAA (69%) were detected in urine. Further analysis of the generated data indicated stool-based PCR and the POC-CCA test to be suitable diagnostics for screening of *S. mansoni* infections, even in the absence of microscopy. A substantial proportion (60%) of the 215 CAA-positive cases showed low antigen concentrations, suggesting that even PCR and POC-CCA underestimated the “true” number of schistosome positives.

## 1. Introduction

Schistosomiasis is a neglected tropical disease affecting over 250 million people worldwide [1], with an estimated 779 million people at risk of the disease [2]. Traditionally, schistosomiasis is diagnosed through microscopical examination of urine (for *Schistosoma haematobium*) or stool (for *S. mansoni*) [3,4]. Although this method is highly specific, it is also known for its low sensitivity—especially in low-intensity infections—leading to underestimation of the prevalence of infection [1]. Alternative and more sensitive diagnostic methods, such as real-time PCR and circulating antigen detection, are progressively used as complementary diagnostics, but these methods do not yet completely replace microscopy. Detecting *Schistosoma* DNA in stool or urine by real-time PCR is proven to be highly specific and more sensitive than microscopy [5,6]. Circulating cathodic antigen (CCA) and circulating anodic antigen (CAA) are two genus-specific carbohydrate antigens that are continuously regurgitated by live *Schistosoma* worms into the bloodstream of the host [7,8], from where they are excreted in the urine [8,9,10] with limited day-to-day variations [11,12]. These characteristics make them excellent markers for detecting active *Schistosoma* infections as well as a proxy for worm burden [13]. A point-of-care test is commercially available for the detection of CCA in urine and is particularly useful for diagnosing intestinal schistosomiasis [14,15,16,17,18,19,20,21] and, to a lesser extent, also for urogenital schistosomiasis [22]. This test has been studied extensively and is currently recommended by the WHO as an alternative to microscopy for diagnosing intestinal schistosomiasis [23,24,25,26]. CAA has a chemically unique structure and is detected using highly sensitive luminescent up-converting reporter particle (UCP) technology in combination with lateral flow (LF) [27]. This laboratory-based UCP-LF CAA test is quantitative and specific for the main human schistosome species (*S. haematobium*, *S. mansoni*, *S. japonicum*, and *S. mekongi*) [27,28,29,30]. The aim of the current study was to investigate the application of a panel of non-microscopy diagnostic methods to determine the presence of *Schistosoma* infections in the absence of microscopy, in order to provide better insight into the performance of these alternative methods as well as to determine whether an accurate diagnosis of schistosomiasis can be made without traditional microscopy.

## 2. Materials and Methods

### 2.1. Study Design and Data Collection

The current study was performed on banked urine and stool samples which were available from a previous study on the prevalence and risk factors of *Taenia solium* cysticercosis conducted in 2009 in Malanga, Bas-Congo, the Democratic Republic of the Congo [31,32]. After obtaining informed consent, participants were asked to provide a stool and urine sample. Due to a lack of time and staff, no extended parasitological examination was performed at the time of sample collection. Samples were stored at 4 °C until transport to the local hospital laboratory, where urine samples were stored at −20 °C. Of each collected stool sample, an aliquot of approximately 1 g was mixed with 2 mL of 70% ethanol and stored at −20 °C. All samples were transported under frozen conditions to the Institute of Tropical Medicine, Antwerp, Belgium, and subsequently transferred to the Leiden University Medical Center (LUMC), Leiden, The Netherlands, and stored at −20 °C until use.

### 2.2. Laboratory Analysis

#### 2.2.1. Real-Time PCR

After DNA extraction, the *Schistosoma* genus-specific real-time PCR was executed as described previously, using a 200 µL sample volume [5,6,32]. *Schistosoma*-specific primers (Ssp48F and Ssp124R) and the double-labeled probe Ssp78T were used to amplify a 77-bp fragment of the internal transcribed spacer-2 (ITS-2) region. An internal control (Phocin Herpes Virus-1) was included for the detection of potential inhibition of amplification. A CFX real-time detection system (Bio-Rad Laboratories, USA) was used for amplification, detection and analysis. The PCR output consisted of a cycle-threshold (Ct)-value, representing the amplification cycle in which the level of fluorescent signal exceeded the background fluorescence and thereby indicating the presence of parasite-specific DNA. Since its implementation, the LUMC-team scored 100% in sensitivity and specificity of their *Schistosoma* PCR at the annual international Helminths External Molecular Assessment Scheme (HEMQAS) provided by the Dutch Foundation for Quality Assessment in Medical Laboratories (SKML) [33]. Intensity of infection was classified arbitrarily as either negative (Ct = 50), low intensity (35 ≤ Ct < 50), medium intensity (30 ≤ Ct < 35), high intensity (25 ≤ Ct < 30), or very high intensity (Ct < 25), based on previous studies [5,34,35].

#### 2.2.2. POC-CCA

The commercially available POC-CCA test (batch no. 50174; Rapid Medical Diagnostics, Pretoria, South Africa) was performed for the detection of CCA, according to the manufacturer’s instructions. In brief, one drop of urine was added to the well of the cassette, followed by one drop of buffer (provided with the test kit). Results were read after 20 min. In case the control line did not develop, the test was considered invalid and the sample was retested. Each POC-CCA cassette was scored as negative, trace (weak line), or positive (1+, 2+, or 3+) by three independent readers, after which the average was taken as the final score. POC-CCA traces were considered negative for the analysis [36].

#### 2.2.3. UCP-LF CAA

The UCP-LF CAA test was performed for the detection of CAA, as described previously [27,37,38]. All urine samples were tested via the UCAA10 format using 10 µL of urine, and subsequently, also with the most sensitive concentration format using 2 mL of urine (UCAA2000). In brief, each urine sample was mixed with an equal volume of 4% trichloroacetic acid, incubated and centrifuged. In the case of the UCAA2000 format, the clear supernatant was concentrated to 20 µL using a 4 mL centrifugal device (Amicon Ultra-4, Millipore, Merck Chemicals B.V., Amsterdam, The Netherlands). The resulting 20 µL concentrate was subsequently used in the assay. Samples with known CAA-levels were included as a reference standard to quantify CAA concentrations as well as to validate the cut-off of the assay. A CAA concentration below 0.1 pg/mL was considered negative [27].

### 2.3. Statistical Analysis

Participants with a complete dataset (i.e., all diagnostic tests performed) were included in the final analysis. Statistical analyses were performed using SPSS version 25 (IBM). Data were summarized using descriptive statistics. The agreement between the diagnostic methods was determined by Kappa (κ) statistics. The nonparametric Spearman’s rank correlation was applied to measure the relationship between PCR Ct-values, POC-CCA scores and CAA-levels. To compare the sensitivity and specificity of the different diagnostic methods, McNemar’s χ^2^ test was used. In the absence of a suitable reference standard, diagnostic accuracy was compared to a composite reference standard (CRS) assuming 100% specificity for PCR as well as for UCP-LF CAA, meaning that an individual was considered positive if PCR and/or UCP-LF CAA was positive.

### 2.4. Ethics Approval and Consent to Participate

Ethical permission for this study was obtained from the Ethical Committee of the University of Kinshasa, DRC, as well as the Institutional Review Board of the Institute of Tropical Medicine in Antwerp, Belgium (No. 650/09) and the Ethical Committee of the University of Antwerp, Belgium (No. 9/11/47). All participants provided written informed consent before the start of the study.

## 3. Results

A complete set of urine and stool samples was available from 314 individuals (46% male, median age 18 years, range 1–80 years). Figure 1 presents an overview of the percentage of positive results of the different diagnostic methods. The highest number of positives was found with the UCP-LF CAA test; in 215 out of 314 (69%) individuals, CAA was detected in urine. The POC-CCA test was positive in 86 (27%) individuals, while 44 (14%) individuals showed a trace. DNA was detected in stool samples of 87 (28%) individuals, while in 10 (3%) individuals, DNA was detected in urine.

### 3.1. Intensity of Infection

The intensity of infection is shown in Table 1. Of the 87 individuals positive by PCR in stool, the majority of infections were of high intensity (58%). Most urine PCR positives were of low to moderate intensity (70%). With the POC-CCA, mainly low-intensity infections were found (56%). The majority of urine CAA positives were of very low to low intensity (61%). Of those individuals positive by UCP-LF CAA only, the median CAA-level was 1.1 pg/mL (range 0.1–298 pg/mL). In Figure 2, the intensity of infection per age group is demonstrated for each diagnostic method. With increasing age, the number of *Schistosoma* DNA positives first increased, peaking in the age group of 15–24 years, and subsequently decreased with increasing age. CCA was detectable in all age groups, but overall a lower prevalence was observed in the oldest age group. Overall, the number of CAA positives was similar in all age groups, except in the youngest (≤5 years) where fewer positives were found. More high-intensity infections were observed in children aged 10–19 years compared to the other age groups.

### 3.2. Diagnostic Accuracy

The agreement between PCR, POC-CCA, and UCP-LF CAA is shown in Figure 3 and Table 2. Of the 314 individuals, 60 (19%) tested positive with all three diagnostic methods, while 228 (73%) individuals tested positive with at least one of the three diagnostic methods. Only one of the diagnostic methods was positive in 125 (40%) individuals; 6 individuals were positive by PCR in stool only, 7 individuals were positive by POC-CCA only, and 112 individuals were positive by UCP-LF CAA only. Of the 10 individuals with detectable DNA in urine, 2 were UCP-LF CAA positive as well, while 8 tested positive with all three additional diagnostic tests. The sensitivity and specificity of the different diagnostic methods compared to the CRS is shown in Table 3. The UCP-LF CAA test showed the highest sensitivity (97%). The PCR in stool and POC-CCA had a comparable sensitivity, 39% and 36%, respectively, while PCR in urine showed a poor sensitivity (5%).

The correlation between DNA-levels in stool and POC-CCA visual scores was strong (Spearman’s rho −0.62, *p* < 0.001); see Figure 4a. A moderate but still significant correlation was observed between the DNA levels in stool and CAA-levels in urine, as well as between CAA-levels in urine and POC-CCA visual scores (Spearman’s rho −0.55, *p* < 0.001 and 0.56, *p* < 0.001, respectively); see Figure 4.

## 4. Discussion

There is a need for a sensitive, specific, rapid and easy to perform diagnostic test for the diagnosis of schistosomiasis. This study is the first to describe the presence of schistosomiasis based on a combination of PCR and circulating antigen detection in the absence of traditional microscopy. It was designed to evaluate different diagnostic tests on banked stool and urine samples in order to provide better insight into the performance of the different tests as well as to determine whether an accurate estimate of the presence of schistosomiasis can be made without traditional microscopy. While a range of studies have evaluated the comparison between microscopy and PCR, or microscopy and circulating antigens, no studies have compared the outcome of real-time PCR on both stool and urine with urine CCA and CAA levels in the same study population.

Our results showed that both the field-applicable POC-CCA as well as the PCR in stool are equally suitable for a first screening of the schistosomiasis prevalence in an endemic region. A fair to moderate agreement was found between the diagnostic methods. Intensity categories of each diagnostic method were either pragmatic (POC-CCA, UCP-LF CAA) or arbitrary (PCR) and therefore difficult to compare, but when looking at an individual level, a positive correlation was observed between the increasing intensity of the POC-CCA visual score and CAA-levels in urine, which corresponds to previous findings [39,40]. Also, a strong correlation was observed between POC-CCA visual scores and Ct values. The majority of cases that were positive by POC-CCA and PCR (in stool) also had detectable CAA-levels in urine, confirming the ability of both tests to detect active *Schistosoma* infections. Still, numerous additional cases, mainly of low intensity (i.e., <10 pg/mL), were detected by UCP-LF CAA, suggesting that the percentage of schistosomiasis positives is much higher than assumed by POC-CCA and PCR alone. Indeed, the UCP-LF CAA test has proven to be an ultra-sensitive test for the detection of active *Schistosoma* infections [40,41].

The high number of cases detected by the diagnostic methods applied in the current study indicate substantial schistosomiasis transmission levels in this study population. The majority of infections are assumed to be caused by *S. mansoni*, based on the higher frequency of DNA present in stool samples compared to urine samples. Although the PCR assay used in this study was not species specific, *Schistosoma* spp. DNA detected in stool samples most likely indicates an infection with *S. mansoni* [5]. This was confirmed by the results of the POC-CCA, which is known to detect mainly *S. mansoni* infections. In this study, very few individuals were urine PCR positive, pointing towards a possible *S. haematobium* infection [6,35]. While these were all confirmed as *Schistosoma* spp positive by the UCP-LF CAA test, 8 out of 10 were also positive by POC-CCA and PCR in stool. This suggests a possible co-infection of *S. haematobium* with *S. mansoni*, although these urine samples might also have been contaminated with stool and, therefore, could represent an *S. mansoni* infection only. Alternatively, ectopic egg elimination cannot be excluded, as *S. mansoni* eggs have been occasionally observed in urine as well [42,43,44]. In such cases, microscopy could have provided additional information concerning the *Schistosoma* species, provided that the infection intensity was sufficiently high.

Urine CAA results showed a similar age-related distribution of *Schistosoma* infection compared to circulating antigen results as well as egg counts from previous studies [45,46]. The prevalence of infection increased with age, but did not decrease in adults, while the intensity of infection decreased in individuals of 20 years and older. Although a high number of positives was observed in school-aged children, still numerous cases were found in children <5 years of age as well as in adults, stressing the importance of improving treatment uptake in these age groups [47]. Furthermore, with increasing age, the presence of CCA and CAA indicates that worms are still present without a relationship with the number of eggs, as indicated by the decreasing number of *Schistosoma* egg DNA positives with increasing age.

While *Schistosoma* species could have been determined with microscopy, we believe it would not have added any extra value here since the detection of DNA in stool as well as CCA in urine (POC-CCA) both point towards the presence of *S. mansoni* infections. Furthermore, the diagnostic methods applied in this study have proven to be more sensitive compared to traditional microscopy, so it is likely that microscopy would have missed several cases due to its limited sensitivity, in particular in detecting low-intensity infections [14,48,49]. Moreover, microscopy is labor-intensive, and the costs are often higher than for example the POC-CCA [50]. Based on consumables only, the costs of real-time PCR and the UCP-LF CAA assay are roughly 10 times higher than POC-CCA. Therefore, in view of the recent recommendation from the WHO, the POC-CCA is considered to be a good alternative for microscopy. However, when more resources are available, real-time PCR and UCP-LF CAA could be considered to obtain a more accurate estimate of the presence of schistosomiasis.

## 5. Conclusions

A moderate to high percentage of *Schistosoma* infections was observed in this study population based on non-microscopy diagnostic methods. The results of this study indicate that the POC-CCA and PCR on stool are suitable screening tools for *S. mansoni* infections when microscopy is unavailable. However, both methods may still significantly underestimate the “true” number of *Schistosoma* infections since a large number of additional, mainly low positive, cases were found by the ultrasensitive and highly specific UCP-LF CAA test. In conclusion, even without microscopy, sufficient alternative diagnostic methods are available to accurately determine the presence as well as the intensity of schistosome infections in an endemic area.

## Figures and Tables

**Figure 1 tropicalmed-07-00315-f001:**
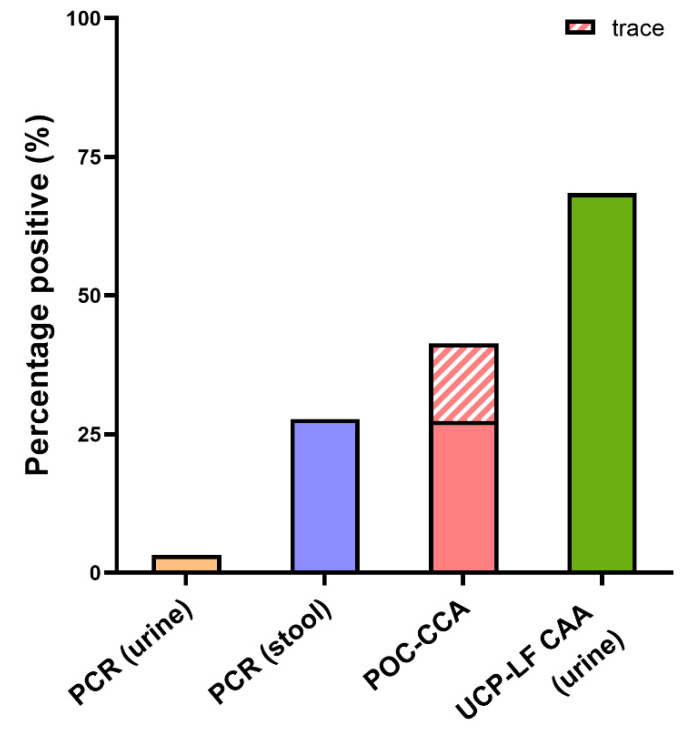
Percentage positive by polymerase chain reaction (PCR) in urine and stool, point-of-care circulating cathodic antigen (POC-CCA), and the up-converting lateral flow circulating anodic antigen (UCP-LF CAA) test in 314 individuals. The shaded area represents POC-CCA trace results.

**Figure 2 tropicalmed-07-00315-f002:**
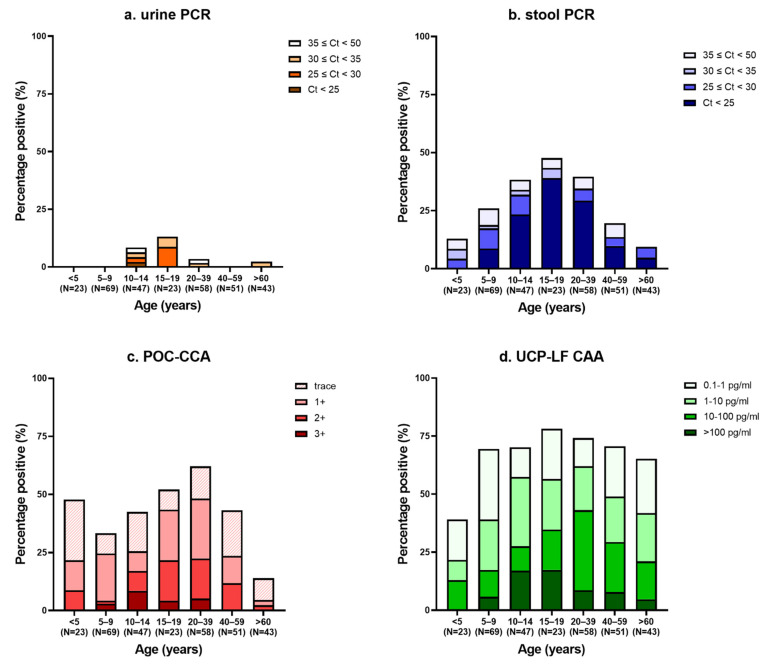
The intensity of infection per age group. Data based on (**a**) PCR in urine; (**b**) PCR in stool; (**c**) Point-of-care circulating cathodic antigen (POC-CCA); and (**d**) up-converting particle lateral flow circulating anodic antigen (UCP-LF CAA).

**Figure 3 tropicalmed-07-00315-f003:**
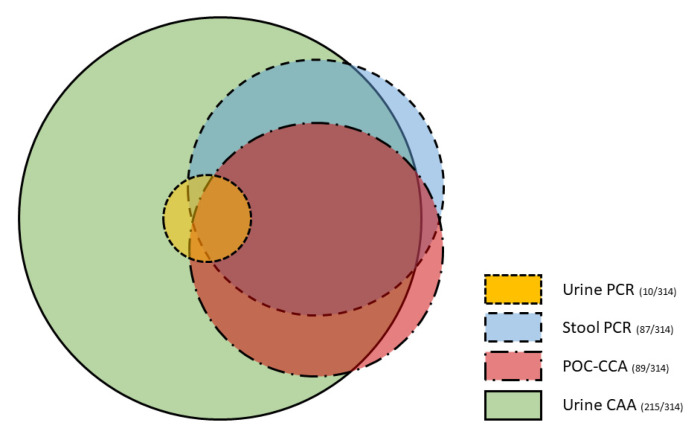
Proportional Venn diagram of polymerase chain reaction (PCR) in urine and stool compared to the point-of-care circulating cathodic antigen (POC-CCA) and the up-converting particle lateral flow circulating anodic antigen (UCP-LF CAA) urine test in 314 individuals.

**Figure 4 tropicalmed-07-00315-f004:**
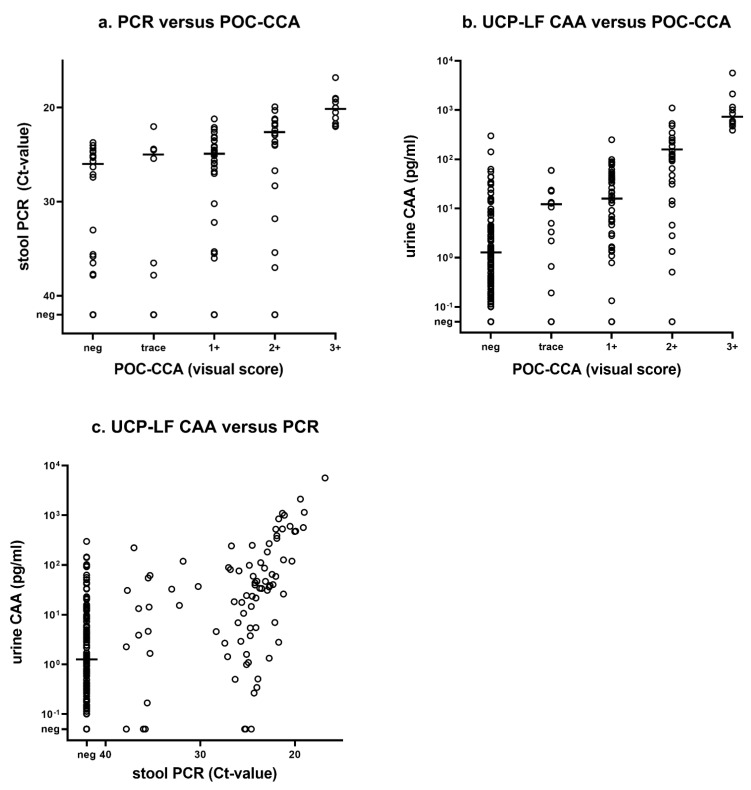
Correlation between polymerase chain reaction (PCR) in stool, point-of-care circulating cathodic antigen (POC-CCA), and the up-converting particle, lateral flow circulating anodic antigen (UCP-LF CAA) urine test: (**a**) PCR versus POC-CCA, (**b**) UCP-LF CAA versus POC-CCA, and (**c**) UCP-LF CAA versus PCR. Horizontal lines indicate the median Ct-value (**a**) or the median CAA concentration (**b**,**c**) of the positive tested samples.

**Table 1 tropicalmed-07-00315-t001:** The intensity of infection based on polymerase chain reaction (PCR) in urine and stool, point-of-care circulating cathodic antigen (POC-CCA), and the up-converting particle, lateral flow circulating anodic antigen (UCP-LF CAA) urine test in 314 individuals.

Diagnostic Method	N (%)
PCR (urine)	
35 ≤ Ct < 50 (low)	2 (0.6%)
30 ≤ Ct < 35 (medium)	4 (1.3%)
25 ≤ Ct < 30 (high)	3 (1.0%)
Ct < 25 (very high)	1 (0.3%)
PCR (stool)	
35 ≤ Ct < 50 (low)	15 (4.8%)
30 ≤ Ct < 35 (medium)	4 (1.3%)
25 ≤ Ct < 30 (high)	18 (5.7%)
Ct < 25 (very high)	50 (15.9%)
POC-CCA	
Trace	44 (14.0%)
1+ (low)	48 (15.3%)
2+ (moderate)	28 (8.9%)
3+ (high)	10 (3.2%)
UCP-LF CAA (urine)	
0.1–1 pg/mL (very low)	64 (20.4%)
1–10 pg/mL (low)	66 (21.0%)
10–100 pg/mL (moderate)	58 (18.5%)
>100 pg/mL (high)	27 (8.6%)

**Table 2 tropicalmed-07-00315-t002:** The level of agreement between polymerase chain reaction (PCR) in urine and stool, point-of-care circulating cathodic antigen (POC-CCA), and the up-converting particle, lateral flow circulating anodic antigen (UCP-LF CAA) urine test by Cohen’s κ coefficient and McNemar’s χ2 test in 314 individuals.

Diagnostic Test	Reference Test		K Value	Interpretation ^1^	*p* Value	McNemar’s *p* Value
	PCR (urine)					
PCR (stool)	Positive	Negative				
Positive	8	79	0.114	Slight	<0.001	<0.001
Negative	2	225				
	POC-CCA					
PCR (stool)	Positive	Negative				
Positive	60	27	0.577	Moderate	<0.001	1
Negative	26	201				
	UCP-LF CAA					
PCR (stool)	Positive	Negative				
Positive	80	7	0.223	Fair	<0.001	<0.001
Negative	135	92				
	UCP-LF CAA					
POC-CCA	Positive	Negative				
Positive	79	7	0.220	Fair	<0.001	<0.001
Negative	136	92				
	PCR (urine)					
POC-CCA	Positive	Negative				
Positive	8	78	0.116	Slight	<0.001	<0.001
Negative	2	226				
	PCR (urine)					
UCP-LF CAA	Positive	Negative				
Positive	10	205	0.030	Slight	0.029	<0.001
Negative	0	99				

^1^ Interpretation of k coefficient: 0, chance; 0.01 to 0.20, slight; 0.21 to 0.40, fair; 0.41 to 0.60, moderate; 0.61 to 0.80, substantial; 0.81 to 0.99, almost perfect.

**Table 3 tropicalmed-07-00315-t003:** Sensitivity and specificity of polymerase chain reaction (PCR) in urine and stool, point-of-care circulating cathodic antigen (POC-CCA), and the up-converting particle circulating anodic antigen (UCP-LF CAA) urine test compared to a composite reference standard (CRS).

		CRS (PCR & UCP-LF CAA) ^1^		Diagnostic Accuracy	
		Positive	Negative	Sensitivity	Specificity
PCR (urine)	Positive	10	0	4.5%	100% ^2^
	Negative	212	92		
PCR (stool)	Positive	87	0	39.2%	100% ^2^
	Negative	135	92		
POC-CCA	Positive	80	6	36.0%	93.5%
	Negative	142	86		
UCP-LF CAA	Positive	215	0	96.8%	100% ^2^
	Negative	7	92		

^1^ Composite reference standard (CRS) was based on PCR (urine/stool) and UCP-LF CAA: an individual was considered positive if at least one of these tests was positive. ^2^ Specificity is 100% by definition.

## Data Availability

Data is contained within the article.

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
