# Peer review of "Diagnosis of Schistosomiasis without a Microscope: Evaluating Circulating Antigen (CCA, CAA) and DNA Detection Methods on Banked Samples of a Community-Based Survey from DR Congo"

_tropicalmed, 2022, doi:10.3390/tropicalmed7100315_

Round 1

Reviewer 1 Report

Please, revise the References section: some journal titles are abbreviated and some are not. Follow the author guidelines of the journal.  

Reviewer 2 Report

In the manuscript “Diagnosis of schistosomiasis without a microscope: a community-based survey using CCA, CAA and DNA detection methods on banked samples from DR Congo” the authors described three different techniques applied to diagnosis schistosomiasis on banked samples. The manuscript is well presented, however material and methods are poorly described. In addition, the title of the manuscript raised a question: in which situation a PCR instrument is available and a microscope not? Economically PCR or circulating antigen technic is more viable than microscopy? This question should be elaborated by the authors in the manuscript, or removed from the title. Follow below other points that should be addressed by the authors.

1. What is CCA and CAA in the title? Remove abbreviations and rewrite the title.

2. Even though the study used samples derivative for a previous study, the authors still should provide the protocol number of the ethic committee authorization for the collection of the samples.

3. Although the methodology of laboratory analysis are referenced, the laboratory analysis methodology should be minimally detailed, specially considering that the study compared those different methods. Authors should designate a methodology subitem for each method evaluated in the study. There are missed information for PCR as reagents, sample quantity and quality measurement and/or concentration, parameters of reaction and reagents primers sequence. Sample processing and instruments for each methodology should be provided as well. All this information should be provided to secure reproducibility of the study.

4. Please, replace the “intensity of infection” for other appropriate term, or simple describe the score results for each technique.

Reviewer 3 Report

The manuscript uses a limited number of banked samples to compare three diagnostic methodologies for Schistosoma eggs in the absence of microscopy. This is a technical manuscript and the title, perhaps, does not reflect it. I think it = is more about the methodologies than about the community-based survey. The authors compared three methodologies with literature support, although only of them (UCP-LF-CAA) is currently by the WHO. The authors also mention that all the methodologies beat microscopy, which defeats the purpose of the study. Indeed, the study presented here supports previous findings. The comparison showed that UCP-LF-CAA outperform the other two methods. At the same time, the results do question the performance of PCR, especially in urine. Without being a specialist in the field, I would be curious to know the amount of analyte that is used for the comparison of the methodologies. How much gDNA from urine and stool is used for the comparison? How does the storage affect the DNA vs. carbohydrate?

Specific comments:

  • Section 3.1 Intensity of Infection. “With increasing age, …) The study focuses on comparing methodologies, this discussion does not fit the objectives/design of the study.
  • “Moreover, microscopy is labour-intensive and the costs are often higher 249 than for example the POC-CCA”. How does it compare to the other two methodologies?
  • The format of the journal in the References section is not consistent.
